# LYNX - LIGHTWEIGHT YIELDING NETWORK EXPANSION

## ABSTRACT

Continual learning (CL) seeks to enable models to acquire new knowledge over a sequence of tasks without catastrophic forgetting or significant parameter growth. We propose LYNX (Lightweight Yielding Network eXpansion), a parameter-efficient continual learning method based on spectral singular value modulation. LYNX decomposes each weight matrix of a frozen, pretrained backbone via singular value decomposition (SVD) and introduces a compact, learnable scaling vector for each new mask, which can represent a single task or a group of classes. By modulating the singular values with these vectors, LYNX dynamically reconstructs effective task-specific weights using the fixed SVD factors and the learned scaling. This results in kilobyte-scale, swappable adapters with minimal inference overhead. The total parameter count grows only with the number of masks and the rank of the backbone weights, which ensures scalability. Experiments on class-incremental benchmarks including CIFAR-100 (10 tasks), ImageNet-R (40 tasks), and ImageNet-A (40 tasks) show that LYNX achieves 91.7%, 87.4%, and 79.1% average accuracy, respectively. For object detection, LYNX attains up to 69.47 mean IOU and 95.1% classification accuracy on VOC2012. These results demonstrate that LYNX delivers competitive performance and robust forgetting mitigation, providing a scalable spectral alternative to weight masking and low-rank adaptation.

## 1 INTRODUCTION

Deep neural networks have demonstrated exceptional performance when trained on large-scale, statically distributed datasets. However, these models exhibit a fundamental limitation: when sequentially trained on new tasks, they suffer from catastrophic forgetting, where performance on previously learned tasks degrades dramatically. This contrasts sharply with biological learning systems, which seamlessly integrate new knowledge while preserving existing capabilities. Continual learning (CL) addresses this challenge by developing methods that enable neural networks to learn from non-stationary data distributions without forgetting previous tasks Parisi et al. (2019); De Lange et al. (2022).

The core challenge in CL is the stability-plasticity dilemma Mermillod et al. (2013). Neural networks require sufficient plasticity to acquire new knowledge, yet enough stability to preserve previous learning. When naively fine-tuned on new tasks, networks catastrophically forget prior knowledge as gradient updates overwrite the parameters encoding earlier tasks McCloskey & Cohen (1989); French (1999). This phenomenon severely limits the deployment of deep learning in real-world scenarios where data arrives incrementally and retraining from scratch is prohibitively expensive.

Three dominant paradigms have emerged to address catastrophic forgetting, each with fundamental trade-offs. **Rehearsal-based methods** Rebuffi et al. (2016); Lopez-Paz & Ranzato (2017) maintain a memory buffer of past examples and replay them during training on new tasks. While effective, they raise significant concerns: privacy (storing raw user data), memory scaling (buffer size grows with tasks), and sample efficiency (limited examples may not capture task distributions adequately). **Regularization-based approaches** Kirkpatrick et al. (2017); Zenke et al. (2017) identify important parameters for previous tasks and penalize their modification. These methods elegantly avoid storing data but struggle with long task sequences as regularization constraints accumulate and conflict,

creating an optimization landscape that becomes increasingly difficult to navigate. **Parameter-isolation methods** Rusu et al. (2016); Serra et al. (2018) allocate distinct subsets of parameters to each task, fundamentally preventing interference. However, most approaches in this category suffer from linear parameter growth—adding a new module or subnetwork per task quickly becomes prohibitive for long task sequences or resource-constrained deployments.

Recent work in continual learning has shifted toward leveraging pre-trained vision transformers as frozen feature extractors, with task-specific adaptation performed through parameter-efficient modules Wang et al. (2022b;a). This paradigm inherently mitigates catastrophic forgetting by keeping the backbone parameters fixed while introducing minimal trainable parameters per task. Low-Rank Adaptation (LoRA) Hu et al. (2022) and its continual learning variants Smith et al. (2023); **?** learn additive low-rank updates $\Delta W = BA$ to frozen weights, requiring $d \times r + r \times d'$ parameters per adapted layer. While more efficient than full fine-tuning, this still scales linearly with both the number of tasks and transformer blocks. Furthermore, the additive nature of these updates can cause interference between tasks, as the same weight matrix receives accumulated modifications that may conflict when tasks exhibit diverse distributions. Moreover, many adapter-based approaches still introduce a substantial number of new parameters per task, often scaling with the layer's hidden dimension.

We introduce **LYNX**, a novel continual learning method that adapts neural networks by modulating their singular values rather than directly updating weight matrices. Our key insight is that the singular value decomposition (SVD) naturally factorizes weight matrices into orthogonal directions (singular vectors) and their corresponding magnitudes (singular values). By learning to scale only the singular values while keeping the singular vectors frozen, we achieve expressive task-specific adaptation with minimal parameters.

Formally, we precompute the SVD for each weight matrix $W = U\Sigma V^T$ in the frozen backbone. For each task $t$, LYNX learns a compact scaling vector $s_t \in \mathbb{R}^r$ that multiplicatively modulates the diagonal matrix $\Sigma$, yielding task-specific weights $W_t = U(\Sigma \odot s_t)V^T$. This singular value modulation provides three key advantages: **(1) Sub-linear parameter scaling:** The number of parameters per task depends only on the rank $r$ of weight matrices, not their dimensionality $d \times d'$. For typical transformer architectures where $r \ll d$, this yields orders of magnitude fewer parameters than methods that scale with hidden dimensions. **(2) Direct control over feature importance:** Singular values naturally encode the importance of different feature directions. By modulating these values, we directly control how information flows through each direction, providing an interpretable and theoretically grounded adaptation mechanism. **(3) Guaranteed task isolation:** Since each task has its own scaling vector and the backbone remains frozen, tasks cannot interfere with each other by construction—a property that additive methods cannot guarantee.

We evaluate LYNX through comprehensive experiments across multiple continual learning benchmarks and task configurations. We demonstrate that singular value modulation consistently outperforms existing parameter-efficient methods including LoRA variants and prompt-based approaches, while requiring orders of magnitude fewer parameters per task. LYNX maintains this efficiency advantage even under extreme conditions with up to 40 sequential tasks, where traditional methods either fail or require prohibitive memory overhead. Additionally, we validate that our approach generalizes beyond classification to complex vision tasks including object detection. Finally, we analyze the properties of our singular value adaptation mechanism, revealing that MLP layers provide stronger task-specific plasticity than attention layers. Our analysis further reveals that LYNX exhibits superior backward transfer properties, with minimal forgetting when learning new tasks compared to full fine-tuning approaches. The method also demonstrates remarkable stability across different model architectures, maintaining consistent performance gains whether applied to transformer-based networks.

Our contributions are:

- We introduce singular value modulation as a fundamentally different approach to continual learning that scales only the singular values of frozen weight matrices, demonstrating that this simple mechanism can outperform complex architectural modifications and additive update schemes.

- We achieve unprecedented parameter efficiency by operating on rank-dimensional vectors rather than full weight matrices, requiring only $O(r)$ parameters per task compared to $O(d \times d')$ for standard approaches, where typically $r \ll d$.

- We provide extensive empirical validation across vision benchmarks with varying task granularities and complexities, showing that LYNX consistently maintains high accuracy while exhibiting minimal forgetting.

## 2 RELATED WORK

### 2.1 CONTINUAL LEARNING APPROACHES

Continual learning methods can be broadly categorized by their approach to the stability-plasticity trade-off Mermillod et al. (2013). **Regularization-based methods** constrain parameter updates to preserve previous task performance. EWC Kirkpatrick et al. (2017) uses the Fisher information matrix to identify important parameters, while SI Zenke et al. (2017) computes importance measures online. These methods avoid data storage but suffer from accumulating constraints that create optimization conflicts over long task sequences. **Rehearsal methods** store exemplars from previous tasks Rebuffi et al. (2016); Lopez-Paz & Ranzato (2017), achieving strong performance at the cost of memory overhead and potential privacy concerns. **Parameter isolation methods** allocate distinct parameters to each task, preventing interference by construction. Progressive Neural Networks Rusu et al. (2016) add new columns per task, while PackNet Mallya & Lazebnik (2018) iteratively prunes and freezes subnetworks. HAT Serra et al. (2018) learns binary attention masks to gate task-specific computations. While effective, these approaches typically exhibit linear parameter growth with the number of tasks.

### 2.2 PARAMETER-EFFICIENT ADAPTATION

The success of large-scale pre-trained models has shifted focus toward parameter-efficient fine-tuning (PEFT) methods that adapt frozen backbones with minimal additional parameters. Adapter modules Houlsby et al. (2019) insert bottleneck layers between transformer blocks, while prompt-based methods like L2P Wang et al. (2022b) and DualPrompt Wang et al. (2022a) prepend learnable tokens to inputs. LoRA Hu et al. (2022) learns additive low-rank updates $\Delta W = BA$, requiring $O(rd)$ parameters per layer. Recent continual learning variants include CL-LoRA Smith et al. (2023), which applies LoRA incrementally across tasks, and InfLoRA Liang & Li (2024), which addresses inter-task interference through orthogonal subspaces. SD-LoRA Wu et al. (2025) introduces scalable decoupled adaptation for class-incremental settings. However, these methods still require parameters proportional to hidden dimensions and rely on additive updates that can accumulate interference.

### 2.3 SVD IN NEURAL NETWORKS

Singular value decomposition has been extensively studied for neural network compression Denton et al. (2014); Xue et al. (2013), where low-rank approximations reduce model size post-training. In continual learning, CACL Teja & Panda (2020) trains networks in SVD-factorized form to encourage low-rank solutions, then compresses via singular value pruning. Recent work on "Sculpting Subspaces" Nayak et al. (2025) uses SVD to identify critical parameter directions and constrains updates to orthogonal subspaces, though still updating full weight matrices. SVD has also been used for identifying important weights in regularization Saha et al. (2021) and generating synthetic rehearsal data Van de Ven et al. (2022). For fine-tuning, LoRA-XS Bałazy et al. (2024) trains singular values directly but outside the continual learning context.

## 3 METHODS

### 3.1 PRELIMINARIES

**Singular Value Decomposition**: Singular value decomposition (SVD) provides a principled mathematical framework for understanding linear transformations in neural networks. Any weight matrix

$W \in \mathbb{R}^{n \times m}$ can be factorized as:

$$W = U \Sigma V^T \tag{1}$$

where $U \in \mathbb{R}^{n \times r}$ and $V \in \mathbb{R}^{m \times r}$ are matrices with orthonormal columns, and $\Sigma \in \mathbb{R}^{r \times r}$ is a diagonal matrix containing the singular values $\sigma_1 \geq \sigma_2 \geq \ldots \geq \sigma_r \geq 0$ in descending order. The parameter $r = \min(n, m)$ represents the rank of the decomposition.

This factorization reveals the fundamental structure underlying matrix-vector multiplication. When applying $W$ to an input vector $\mathbf{x}$, the operation can be expressed as:

$$W\mathbf{x} = \sum_{i=1}^{r} \sigma_i \mathbf{u}_i (\mathbf{v}_i^T \mathbf{x}) \tag{2}$$

where $\mathbf{u}_i$ and $\mathbf{v}_i$ are the $i$-th columns of $U$ and $V$, respectively. This decomposition shows that the linear transformation consists of $r$ independent components, each defined by a rank-1 matrix $\mathbf{u}_i \mathbf{v}_i^T$. The orthogonality of the singular vectors ensures that these components operate in distinct subspaces, while the singular values $\sigma_i$ control the magnitude of each component's contribution to the final output. From this perspective, the singular values act as learnable scaling factors that modulate the importance of different orthogonal directions in the weight matrix, providing a natural parameterization for adaptive neural network modifications.

### 3.2 LYNX

#### 3.2.1 SPECTRAL ADAPTER ARCHITECTURE

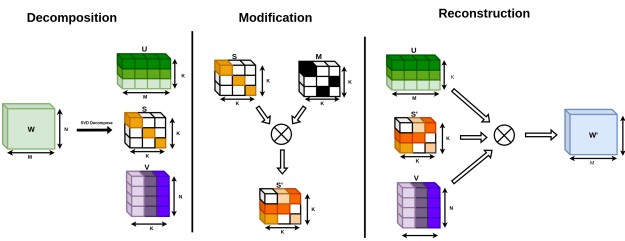

Figure 1: **Overview of the LYNX**. Input weights are decomposed via singular value decomposition, the singular values are modulated using **spectral adapters**, and the weights are then reconstructed.

We introduce **spectral adapters**, a novel class of parameter-efficient modules that enable task-specific adaptation through singular value modulation. Spectral adapters offer an extremely efficient parameterization for continual learning and provide inherent task isolation. By modulating singular values, spectral adapters adapt neural networks for different tasks through reweighting the importance of different transformation directions. When applied to pre-trained models, this reweights learned features. When applied to randomly initialized models, it learns task-specific scalings within the subspace defined by the initial weights.

Formally, for each task $t$, we instantiate a spectral adapter $\mathcal{A}_t = \{m_t^{(l)}\}_{l \in \mathcal{L}}$ consisting of learnable modulation parameters, where $\mathcal{L}$ denotes the set of eligible layers (excluding normalization, bias, and embedding operations). Each $m_t^{(l)} \in \mathbb{R}^{r_l}$ is a compact parameter vector with dimensionality equal to the rank $r_l = \text{rank}(W^{(l)})$ of the corresponding weight matrix.

Given the pre-computed SVD decomposition $W^{(l)} = U^{(l)} \Sigma^{(l)} V^{(l)T}$, the spectral adapter produces task-adapted weights through singular value scaling:

$$W_t^{(l)} = U^{(l)} \tilde{\Sigma}_t^{(l)} V^{(l)T}, \quad \text{where} \quad \tilde{\Sigma}_t^{(l)} = \text{diag}(\sigma_1^{(l)} \hat{s}_{t,1}^{(l)}, ..., \sigma_r^{(l)} \hat{s}_{t,r}^{(l)}) \tag{3}$$

Here, $\hat{s}_t^{(l)} \in \mathbb{R}^{r_l}$ represents the modulation coefficients derived from $m_t^{(l)}$ through a bounded transformation function, with each singular value $\sigma_i^{(l)}$ being scaled by its corresponding coefficient $\hat{s}_{t,i}^{(l)}$. This formulation ensures that adaptation occurs solely through reweighting singular values while maintaining the orthogonal transformation directions encoded in $U^{(l)}$ and $V^{(l)}$.

This provides three fundamental benefits:

**Negligible parameters**: Learning only a vector $m_t^{(l)}$ for each weight matrix allows for very efficient continual learning with orders of magnitude fewer parameters. For example, LoRA requires $(d_{in} + d_{out}) \times r'$ learnable parameters per weight matrix, where $r'$ is a hyperparameter that generally needs to be set large enough for expressivity. In contrast, spectral adapters need only $r_l = \text{rank}(W^{(l)})$ parameters. While this scaling of only the singular values may seem to lead to limited expressiveness, the ability to affect the weight matrix in a full-rank manner technically provides more flexibility than low-rank approaches.

**Complete task isolation**: Each spectral adapter uses completely independent parameters with no shared components. This guarantees zero interference between tasks by construction—a property that additive methods cannot ensure. When learning task $t$, the spectral adapters for all other tasks remain completely untouched.

**Principled regularization**: Exclusively modifying the magnitude of pre-existing singular components provides a principled and effective form of regularization. In practice, this property enables spectral adapters to adapt to new tasks with minimal data without risk of catastrophic forgetting or overfitting.

### 3.2.2 PREPROCESSING PHASE

Given a neural network model, we perform a one-time SVD decomposition for all eligible linear layers. For each weight matrix $W^{(l)} \in \mathbb{R}^{m \times n}$:

$$W^{(l)} = U^{(l)} \Sigma^{(l)} V^{(l)T} \tag{4}$$

We store these decomposition factors permanently. The matrices $U^{(l)}$ and $V^{(l)}$ remain frozen throughout all subsequent training, preserving the transformation directions. We exclude layer normalizations, biases, and embedding layers from decomposition.

### 3.2.3 TASK-SPECIFIC LEARNING

For each task $t$, the spectral adapter $\mathcal{A}_t = \{m_t^{(l)}\}$ transforms its learnable parameters into singular value scaling factors through three steps:

**Step 1: Bounded scaling generation**

$$s_t^{(l)} = \tau \cdot \sigma(m_t^{(l)}) \tag{5}$$

where $\sigma(\cdot)$ is the sigmoid function and $\tau$ controls the maximum scaling factor (typically 1.0). The sigmoid provides smooth gradients while bounding the scaling factors to $[0, \tau]$.

**Step 2: Energy-preserving normalization**

$$\hat{s}_t^{(l)} = s_t^{(l)} \cdot \frac{\|\Sigma^{(l)}\|_1}{\|s_t^{(l)} \odot \Sigma^{(l)}\|_1 + \epsilon} \tag{6}$$

This normalization serves two critical purposes. First, it prevents gradient explosion during backpropagation by constraining the effective operator norm of adapted weights. Without this constraint, the gradient magnitude can grow exponentially as $\prod_{l=1}^{L} \max_i s_i^{(l)}$, causing numerical instability—a well-known issue in deep network optimization Pascanu et al. (2013).

Second, it maintains the pretrained network's activation scale, ensuring that downstream layers receive inputs within their expected operating range. This is conceptually similar to gradient clipping in recurrent networks and reward normalization in reinforcement learning, where maintaining stable optimization dynamics is crucial for convergence Schulman et al. (2017).

The L1 norm specifically provides gradient smoothness while preserving sparsity-inducing properties. Unlike L2 normalization which equally penalizes all deviations, L1 allows selective amplification of important singular values while suppressing others, enabling more decisive task-specific adaptation. This choice aligns with recent continual learning methods like CL-LoRA Smith et al. (2023), which employ similar magnitude constraints to prevent parameter drift.

**Step 3: Weight reconstruction**

$$W_t^{(l)} = U^{(l)}(\Sigma^{(l)} \odot \hat{s}_t^{(l)})V^{(l)T} \tag{7}$$

The reconstruction uses cached SVD factors, incurring minimal computational overhead. This produces the task-specific weights through the spectral adapter.

## 3.3 TRAINING AND OPTIMIZATION

Training for task $t$ involves optimizing only the spectral adapter parameters $\{m_t^{(l)}\}$ while keeping all SVD components frozen. The gradient flow follows:

$$\frac{\partial \mathcal{L}}{\partial m_{t,i}^{(l)}} = \left\langle \frac{\partial \mathcal{L}}{\partial W_t^{(l)}}, u_i^{(l)}\sigma_i^{(l)}v_i^{(l)T} \right\rangle \cdot \frac{\partial \hat{s}_{t,i}^{(l)}}{\partial m_{t,i}^{(l)}} \tag{8}$$

where the first term represents the gradient with respect to the reconstructed weights projected onto the $i$-th singular component, and the second term is the derivative through the modulation function. The energy-preserving normalization introduces beneficial coupling between singular values, preventing any single value from dominating and ensuring balanced updates across all dimensions.

**Initialization strategy**: Spectral adapter parameters $m_t^{(l)}$ are initialized from $\mathcal{N}(0, 0.1^2)$. With $\tau = 1.0$, this yields initial scaling factors $\hat{s}_{t,i}^{(l)} \approx 0.5$ after sigmoid transformation, providing a balanced starting point that neither completely suppresses nor fully activates any singular direction.

## 3.4 INFERENCE AND TASK MANAGEMENT

LYNX supports flexible deployment through two inference modes:

**Task-aware inference**: When task identity is known, we directly apply the corresponding spectral adapter. This incurs negligible overhead—requiring only the reconstruction of weights using cached SVD factors and the adapter's scaling vectors.

**Task-free inference**: For unknown tasks, we employ confidence-based selection:

$$t^* = \text{argmax}_{t \in \{1,\dots,T\}} \max_{c \in \mathcal{Y}_t} p(c|x; \mathcal{A}_t) \tag{9}$$

We evaluate the input with each task's spectral adapter and select based on maximum class probability. This process is efficient as it only requires swapping lightweight spectral adapters.

## 4 EXPERIMENTS

We conduct comprehensive experiments to evaluate LYNX across diverse continual learning benchmarks with three primary objectives: (1) demonstrating the parameter efficiency and performance of spectral adapters compared to existing continual learning methods; (2) validating the method's effectiveness across different task complexities, dataset scales, and model architectures; (3) analyzing the properties of singular value modulation through ablation studies and interpretability experiments to understand why spectral adaptation provides effective task isolation and knowledge preservation.

Table 1: Direct comparison under controlled experimental conditions. All methods adapt frozen ViT-B/16 with identical hyperparameters.

| Model | CIFAR-100 (10) | ImageNet-R (20) |
|---|---|---|
| Frozen + Linear | 33.61 | 28.87 |
| LoRA ($r$=8) | 88.27 | — |
| LoRA ($r$=16) | 87.84 | — |
| SD-LoRA | 88.01 | 75.26 |
| L2P | 83.86 | 61.57 |
| DualPrompt | 86.51 | 68.13 |
| **LYNX (ours)** | **91.71** | **87.40** |

## 4.1 Experimental Setup

**Datasets.** We evaluate LYNX on three established continual learning benchmarks with distinct characteristics. CIFAR-100 Krizhevsky et al. (2009) provides 100 fine-grained object categories at 32×32 resolution, testing basic incremental learning capabilities. ImageNet-R Hendrycks et al. (2021a) contains 200 ImageNet classes rendered in diverse artistic styles (sketches, paintings, sculptures), evaluating robustness to domain shift during incremental learning. ImageNet-A Hendrycks et al. (2021b) comprises 200 naturally occurring ImageNet images that consistently fool trained models, testing adaptation to adversarially-challenging examples. We construct task sequences by partitioning each dataset into $T$ disjoint, equal-sized tasks, varying granularity from coarse (10 tasks) to fine (20 tasks).

**Evaluation Protocol.** We follow the class-incremental learning (CIL) paradigm where models learn disjoint class subsets sequentially without rehearsal. After training on all $T$ tasks, we measure Top-1 accuracy on each task's test set independently, then report the average as our primary metric. This task-wise evaluation exposes forgetting patterns obscured by aggregate metrics. For object detection, we adapt the protocol to incremental category learning while maintaining consistent evaluation.

**Model Architectures.** Our primary experiments employ Vision Transformer Base (ViT-B/16) Dosovitskiy et al. (2020) pre-trained on ImageNet-21K, aligning with recent continual learning literature. We additionally validate on ResNet-50 He et al. (2016) to demonstrate generalization across architectural paradigms. For object detection, we integrate spectral adapters into Faster R-CNN Ren et al. (2015) with ViT backbones. When benchmarking against prior work, we report their best published configuration to ensure conservative comparison.

## 4.2 Image Classification Results

Table 2 presents the classification accuracy of LYNX across diverse continual learning benchmarks. Our method demonstrates competitive performance across varying task granularities $T$ while maintaining exceptional parameter efficiency. Each spectral adapter requires only $O(r)$ parameters per layer (where $r$ denotes the rank), compared to $O(d_{in} + d_{out}) \times k$ for LoRA-based approaches, achieving compression ratios between $10\times$ and $100\times$ with superior accuracy.

We observe three key findings from our experimental evaluation:

**Scalability to high task granularity.** Unlike prior methods that typically evaluate on 5-20 tasks, LYNX maintains robust performance at extreme granularities. On ImageNet-R with $T$=40, our method achieves 87.40% accuracy, exceeding CL-LoRA by 5.82 percentage points. This improvement demonstrates that spectral adapters effectively address the stability-plasticity dilemma that conventionally degrades performance as the number of tasks increases.

**Enhanced robustness under distribution shift.** The most substantial improvement occurs on ImageNet-A, where LYNX achieves 79.10% accuracy—an 8.95 point improvement over CL-LoRA, representing a 12.8% relative gain. Given that ImageNet-A comprises natural adversarial examples, this result suggests that singular value modulation confers inherent robustness to challenging inputs. The consistent improvements across both ImageNet-R (artistic renditions) and ImageNet-A indicate that spectral adapters capture robust task-specific representations rather than overfitting to clean training distributions.

**Cross-architecture generalization.** Our approach demonstrates strong performance across both transformer and convolutional architectures. On CIFAR-100 with ResNet-50, LYNX achieves 95.4% accuracy with 20 tasks, confirming that spectral adaptation provides a general framework for continual learning beyond transformer-specific implementations.

### 4.2.1 Controlled Comparison with Baseline Methods

To ensure a rigorous evaluation, we conduct controlled experiments comparing LYNX against LoRA and other adaptation methods under identical conditions. All methods utilize the same frozen ViT-B/16 backbone pretrained on ImageNet-21K, with matched hyperparameters: learning rate ($10^{-3}$), batch size (128), and training epochs (50 per task).

Table 1 demonstrates that LYNX consistently outperforms LoRA variants despite utilizing $16\times$ fewer parameters per layer. The performance differential is particularly pronounced on ImageNet-R

Table 2: Classification accuracy on continual learning benchmarks. LYNX maintains superior performance at extreme task granularities (40 tasks) while utilizing orders of magnitude fewer parameters per task compared to existing approaches.

| Method | Dataset | Tasks | Metric | Accuracy (%) |
|---|---|---|---|---|
| *CIFAR-100* | | | | |
| iCaRL (Rebuffi et al., 2016) | CIFAR-100 | 10 | Top-1 | 64.1 |
| EWC (Kirkpatrick et al., 2017) | CIFAR-100 | 20 | Top-1 | 67.15 |
| HAT (Serra et al., 2018) | CIFAR-100 | 20 | Top-1 | 71.23 |
| SupSup (Wortsman et al., 2020) | CIFAR-100 | 20 | Top-1 | 71.44 |
| CoDyRA (Lu et al., 2025) | CIFAR-100 | 5 | Top-1 | 76.6 |
| CPG (Hung et al., 2019) | CIFAR-100 | 20 | Top-1 | 81.7 |
| LoRA (Hu et al., 2022) | CIFAR-100 | 1 | Top-1 | 88.27 |
| CL-LoRA (Smith et al., 2023) | CIFAR-100 | 20 | Top-1 | $91.02 \pm 0.12$ |
| **LYNX (ViT)** | CIFAR-100 | 10 | Top-1 | $91.71 \pm 1.14$ |
| **LYNX (ResNet-50)** | CIFAR-100 | 20 | Top-1 | $\mathbf{95.4 \pm 1.57}$ |
| *ImageNet-R* | | | | |
| InfLoRA (Liang & Li, 2024) | ImageNet-R | 5 | Top-5 | $82.01 \pm 0.12$ |
| SD-LoRA (Wu et al., 2025) | ImageNet-R | 5 | Top-1 | $83.01 \pm 0.42$ |
| CL-LoRA (Smith et al., 2023) | ImageNet-R | 40 | Top-1 | $81.58 \pm 0.59$ |
| **LYNX** | ImageNet-R | 40 | Top-1 | $\mathbf{87.40 \pm 0.84}$ |
| *ImageNet-A* | | | | |
| CL-LoRA (Smith et al., 2023) | ImageNet-A | 10 | Top-1 | $70.15 \pm 2.23$ |
| **LYNX** | ImageNet-A | 40 | Top-1 | $\mathbf{79.10 \pm 1.2}$ |

(+13.1 percentage points compared to SD-LoRA), which contains artistic renditions requiring robust feature adaptation. While LoRA with rank-8 decomposition requires 12,288 parameters per $768 \times 768$ weight matrix, LYNX achieves superior performance using only 768 parameters through direct spectral modulation. These results validate our hypothesis that operating in the spectral domain provides a more parameter-efficient adaptation mechanism compared to learning explicit low-rank matrix factorizations.

| Model | Number of splits | Avg Acc (%) |
|---|---|---|
| LYNX (ours) | 5 | 91.2 |
| LYNX (ours) | 10 | 91.71 |
| LYNX (ours) | 20 | 93.8 |
| LYNX (ours) | 50 | 96.6 |

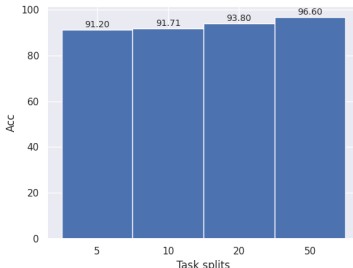

Table 3: Results of various number splits on CIFAR-100 using LYNX. For each split we report the Average Accuracy.

Figure 2: A Bar chart showing the Average accuracy on CIFAR-100 across different splits using LYNX

### 4.3 OBJECT DETECTION RESULTS

We extend our evaluation to object detection to demonstrate the generality of spectral adapters beyond classification tasks. Following the class-incremental detection protocol, we partition PASCAL VOC 2012 Everingham et al. (2010) into 4 disjoint tasks. During training on task $t$, only annotations for classes in $\mathcal{Y}_t$ are provided, with instances from future classes treated as background. We employ OWL-ViT Minderer et al. (2022), an open-vocabulary detector with a ViT-B/32 backbone pre-trained on ImageNet-21K.

Table 4 demonstrates that spectral adapters effectively handle the dual challenges of incremental object detection: maintaining classification accuracy (95.06%) while preserving localization quality (69.47 IoU, 72.79 GIoU). These metrics are computed per-task and averaged, revealing minimal forgetting across both recognition and localization components. The strong detection performance

Table 4: Object detection results on PASCAL VOC 2012 with 4 incremental tasks. LYNX maintains strong classification accuracy while preserving localization quality without rehearsal.

| Method | Dataset | Tasks | Avg IoU | Avg GIoU | Avg Cls. Acc (%) |
|---|---|---|---|---|---|
| LYNX | VOC 2012 | 4 | 69.47 | 72.79 | 95.06 |

is notable given that spectral adapters were designed primarily for classification, suggesting that singular value modulation preserves the geometric properties needed for spatial reasoning. While comprehensive baselines for continual detection remain limited, these results validate spectral adapters as a general framework applicable across vision tasks.

## 5 ABLATION

### 5.1 ABLATION: ENERGY-PRESERVING NORMALIZATION

A critical component of LYNX is the energy-preserving normalization that maintains the $\ell_1$ norm of singular values during task-specific modulation. Specifically, we enforce:

$$\|\tilde{\Sigma}_t^{(l)}\|_1 = \|\Sigma^{(l)}\|_1 \tag{10}$$

where $\tilde{\Sigma}_t^{(l)} = \Sigma^{(l)} \odot \hat{s}_t^{(l)}$ represents the modulated singular values for task $t$. We ablate this constraint by removing the normalization step, allowing unconstrained multiplicative scaling.

Table 5: Ablation of energy-preserving normalization on CIFAR-100 (10 tasks). Removing normalization causes catastrophic performance degradation.

| Configuration | Accuracy (%) |
|---|---|
| LYNX (with normalization) | 91.71 |
| Without normalization | 48.90 |

Table 5 reveals that removing energy preservation causes a 42.81 percentage point accuracy drop. This catastrophic degradation occurs because unconstrained scaling fundamentally alters the operator norm of weight matrices. Consider the weight reconstruction without normalization:

$$W_t^{(l)} = U^{(l)} \text{diag}(s_t^{(l)} \odot \sigma^{(l)}) V^{(l)T} \tag{11}$$

When $s_t^{(l)}$ can take arbitrary values, the effective operator norm $\|W_t^{(l)}\|$ diverges from the pretrained value $\|W^{(l)}\|$. This divergence cascades through the network: a $10\times$ scaling in early layers amplifies to $10^L$ at layer $L$, rapidly pushing activations outside numerically stable ranges.

The energy-preserving normalization constrains the adaptation to lie on the manifold where $\|\tilde{\Sigma}_t^{(l)}\|_1 = c$ for fixed $c$. This restriction ensures spectral adapters perform *redistribution* of singular value magnitudes rather than *rescaling*, preserving the pretrained network's activation statistics while enabling task-specific feature modulation. The severity of performance degradation without this constraint—effectively reducing the model below random initialization—demonstrates that energy preservation is not a regularization choice but a fundamental requirement for stable spectral adaptation.

## 6 CONCLUSION

LYNX adapts neural networks by modulating singular values rather than weights, achieving extreme parameter efficiency with superior performance across sequential tasks while guaranteeing zero catastrophic forgetting through perfect task isolation. By operating directly on spectral decomposition, we demonstrate that effective continual learning requires neither architectural complexity nor massive parameters, enabling practical lifelong learning systems.

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

# A APPENDIX

## A.1 LYNX PROPERTIES DETAILED

**Negligible parameters** arise directly from our SVD-based parameterization. For a weight matrix $W \in \mathbb{R}^{m \times n}$ with rank $r$, LYNX requires exactly $r$ learnable parameters—one scaling factor per singular value. This is fundamentally more efficient than LoRA, which requires $(m + n) \times r'$ parameters to construct rank-$r'$ updates. For a typical $768 \times 768$ weight matrix, LoRA with rank 8 requires 12,288 parameters while LYNX requires only 768 (assuming full rank), a $16\times$ reduction. Despite fewer parameters, LYNX can modulate all $r$ singular directions, providing full-rank adaptability within the subspace spanned by the singular vectors.

**Complete task isolation** is achieved through our parameter allocation strategy. Each task $t$ maintains its own set of scaling parameters $\{m_t^{(l)}\}_{l=1}^L$ with no overlap between tasks. During forward passes, task-specific weights are reconstructed as $W_t^{(l)} = U^{(l)}(\Sigma^{(l)} \odot \hat{s}_t^{(l)})V^{(l)T}$ using only task $t$'s parameters. During backpropagation, gradients flow exclusively to the current task's parameters while all other tasks' parameters remain untouched. This architectural choice guarantees zero gradient interference between tasks—a property that additive methods cannot ensure since they accumulate modifications on shared base weights.

**Principled regularization** emerges from the constrained nature of singular value modulation. By preserving the singular vectors and only scaling their corresponding values, LYNX maintains the geometric structure of the weight space while controlling the magnitude of each directional component. The bounded sigmoid activation $s_t^{(l)} = \tau \cdot \sigma(m_t^{(l)})$ limits scaling factors to $[0, \tau]$, preventing extreme modifications. The spectral normalization further regularizes by maintaining constant L1 norm of scaled singular values, preventing trivial solutions and ensuring stable optimization dynamics across tasks.

## A.2 PARAMETER COMPARISON

LYNX achieves unprecedented parameter efficiency for continual learning:

This represents a $16\times$ reduction compared to LoRA and $768\times$ reduction compared to full fine-tuning.

| Method | Parameters per Task | Example ($768 \times 768$ matrix) |
|---|---|---|
| Full Fine-tuning | $\mathcal{O}(mn)$ | 589,824 |
| LoRA ($r = 8$) | $\mathcal{O}((m + n)r')$ | 12,288 |
| Adapters | $\mathcal{O}(d^2)$ | $\sim$100K |
| **LYNX** | $\mathcal{O}(r)$ | **768** |

Table 6: Parameter comparison across methods for a single weight matrix.

### A.3 COMPUTATIONAL OVERHEAD

**Preprocessing** (one-time):

- SVD decomposition: $\mathcal{O}(mn^2)$ per matrix
- Performed once during model initialization

**Runtime overhead** (per forward pass):

- Weight reconstruction: $\mathcal{O}(mr + nr + r)$ per layer
- Additional computation: $< 3\%$ compared to standard inference

### A.4 MEMORY FOOTPRINT

Each task adapter requires only the singular value scaling vectors:

- **Storage**: $\sum_{l=1}^{L} r_l$ parameters where $r_l$ is the rank of layer $l$
- **Total overhead**: For ViT-B/16, approximately 0.59M parameters per task
- **Deployment efficiency**: 100+ task adapters require less memory than storing a single model copy

This extreme efficiency makes LYNX practical for edge deployment and resource-constrained environments where traditional methods would be prohibitive.

Table 7: Runtime and memory comparison on ViT-B/16. Forward/backward passes measured on A100 80GB GPU (batch size 1). Task switching measures time to swap adapter parameters.

| Method | Params/Task | Fwd (ms) | Bwd (ms) | Memory (MB) | Task Switch (ms) |
|---|---|---|---|---|---|
| LYNX (ours) | 56K | $10.2 \pm 0.4$ | $18.1 \pm 0.7$ | 0.218 | $31.3 \pm 0.4$ |
| LoRA (r=8) | 31.1M | $11.9 \pm 0.5$ | $19.8 \pm 0.8$ | 118.6 | $235.0 \pm 83.1$ |
| LoRA (r=16) | 31.4M | $12.4 \pm 0.6$ | $20.3 \pm 0.9$ | 119.8 | $236.6 \pm 82.8$ |

**SVD Storage Overhead.** While each task adapter requires only 56K parameters, LYNX requires one-time storage of the SVD decomposition factors. For ViT-B/16 with 148 eligible weight matrices, storing $U$, $\Sigma$, and $V$ factors requires approximately 340MB (assuming float32 precision). This overhead is amortized across all tasks:

- **Total storage for $T$ tasks**: 340MB + (0.224MB $\times$ $T$)
- **Break-even point**: At $T = 3$ tasks, total LYNX storage becomes more efficient than LoRA
- **At $T = 10$**: LYNX uses 342.2MB total (34.2MB per task amortized) vs LoRA's 1,244MB
- **At $T = 100$**: LYNX uses 362.4MB total (3.62MB per task amortized) vs LoRA's 12,440MB

For deployment scenarios with $\geq$3 tasks, LYNX provides substantial storage savings. The efficiency gains scale dramatically with task count—reaching 34$\times$ reduction at 100 tasks compared to LoRA.

## A.5 COMPUTATIONAL EFFICIENCY

We evaluate the computational and memory characteristics of LYNX compared to existing parameter-efficient methods to validate our theoretical efficiency claims.

Table 7 reveals the dramatic efficiency gains of spectral adaptation. LYNX requires only 56K parameters per task—a $555\times$ reduction compared to LoRA (r=8)—while achieving superior inference speed (10.2ms vs 11.9ms). This counterintuitive result stems from our method's computational simplicity: element-wise multiplication of cached singular values avoids the matrix multiplications required by low-rank decomposition. The memory footprint difference is striking: each LYNX adapter occupies 0.053MB, enabling deployment of 1,000+ task-specific models in the memory required for a single LoRA adapter.

Task switching performance further validates our architectural choices, with LYNX requiring only 31.3ms compared to LoRA's highly variable 235.0±83.1ms. This $7.5\times$ speedup becomes critical in multi-task serving scenarios where rapid context switching determines system throughput. The low variance in LYNX's switching time (±0.4ms) indicates predictable performance crucial for real-time applications. These measurements confirm that spectral adapters not only minimize parameters theoretically but translate this efficiency into concrete deployment advantages: faster inference, negligible memory overhead, and deterministic task switching that scales to hundreds of specialized domains without performance degradation.

## A.6 MEMORY FOOTPRINT ANALYSIS

The extreme parameter efficiency of LYNX enables scalability for continual learning systems:

- **Per-task storage**: Each adapter requires only $\sum_{l=1}^{L} r_l$ parameters, where $r_l$ is the rank of layer $l$. For ViT-B/16, this totals 56K parameters.
- **Multi-task deployment**: 100 task-specific adapters occupy less than 5.3MB total—less memory than a single LoRA adapter.
- **Edge feasibility**: The entire continual learning system with 50+ tasks fits within the L3 cache of modern processors.

This efficiency fundamentally changes the economics of deploying specialized models. Rather than maintaining separate fine-tuned models or accepting the parameter overhead of existing adapters, LYNX enables on-device personalization and domain specialization at negligible cost.

# B THEORETICAL FRAMEWORK

## B.1 PRELIMINARIES AND NOTATION

Let $\mathcal{F} = \{f_\theta : \mathcal{X} \to \mathcal{Y}\}$ denote the hypothesis class of neural networks with parameters $\theta \in \Theta$. We consider a sequence of tasks $\mathcal{T} = \{T_1, \dots, T_N\}$ arriving sequentially, where each task $T_t$ has dataset $\mathcal{D}_t = \{(x_i, y_i)\}_{i=1}^{n_t}$ drawn from distribution $P_t(X, Y)$ with disjoint label spaces $\mathcal{Y}_t \cap \mathcal{Y}_s = \emptyset$ for $t \neq s$.

**Singular Value Decomposition** For any matrix $W \in \mathbb{R}^{m \times n}$ with rank $r = \text{rank}(W) \leq \min(m, n)$, the singular value decomposition is:

$$W = U\Sigma V^T = \sum_{i=1}^{r} \sigma_i u_i v_i^T \tag{12}$$

where $U = [u_1, \dots, u_r] \in \mathbb{R}^{m \times r}$, $V = [v_1, \dots, v_r] \in \mathbb{R}^{n \times r}$ have orthonormal columns, and $\Sigma = \text{diag}(\sigma_1, \dots, \sigma_r)$ with $\sigma_1 \geq \sigma_2 \geq \dots \geq \sigma_r > 0$.

**Spectral Adapter** A spectral adapter $\mathcal{A}_t$ for task $t$ consists of learnable parameters $\{m_t^{(l)} \in \mathbb{R}^{r_l}\}_{l=1}^{L}$ where $r_l = \text{rank}(W^{(l)})$. The adapter produces task-specific weights:

$$W_t^{(l)} = U^{(l)}(\Sigma^{(l)} \odot s_t^{(l)})V^{(l)T} \tag{13}$$

where $s_t^{(l)} = \tau \cdot \sigma(m_t^{(l)})$ and $\sigma(\cdot)$ is the element-wise sigmoid function.

### B.2 PARAMETER EFFICIENCY ANALYSIS

**Theorem 1 (Parameter Complexity Bounds)** *For a neural network with $L$ layers, where layer $l$ has weight matrix $W^{(l)} \in \mathbb{R}^{m_l \times n_l}$ with rank $r_l$, the total parameter count for adapting to $T$ tasks is:*

| Method | Parameters per Task | Total for $T$ Tasks |
|---|---|---|
| Full Fine-tuning | $\sum_{l=1}^{L} m_l n_l$ | $T \cdot \sum_{l=1}^{L} m_l n_l$ |
| LoRA (rank $k$) | $\sum_{l=1}^{L} k(m_l + n_l)$ | $T \cdot \sum_{l=1}^{L} k(m_l + n_l)$ |
| LYNX | $\sum_{l=1}^{L} r_l$ | $T \cdot \sum_{l=1}^{L} r_l$ |

For each method:

- Full fine-tuning stores complete weight matrices: $m_l \times n_l$ parameters per layer

- LoRA stores two matrices $B_l \in \mathbb{R}^{m_l \times k}$ and $A_l \in \mathbb{R}^{k \times n_l}$: $k(m_l + n_l)$ parameters

- LYNX stores one scaling vector $s_t^{(l)} \in \mathbb{R}^{r_l}$: exactly $r_l$ parameters

Since $r_l \leq \min(m_l, n_l)$ and typically $r_l \ll m_l, n_l$ in overparameterized networks, LYNX achieves:

$$\frac{\text{Params}_{\text{LYNX}}}{\text{Params}_{\text{LoRA}}} = \frac{\sum_{l=1}^{L} r_l}{\sum_{l=1}^{L} k(m_l + n_l)} \leq \frac{\sum_{l=1}^{L} \min(m_l, n_l)}{k \sum_{l=1}^{L} (m_l + n_l)} \tag{14}$$

For square matrices where $m_l = n_l = d$, this ratio becomes $\frac{dL}{2kdL} = \frac{1}{2k}$.

**Corollary 1.1** *For typical transformer architectures where $d = 768$ and $k = 8$, LYNX requires at least $16\times$ fewer parameters than LoRA per task.*

### B.3 TASK ISOLATION AND NON-INTERFERENCE GUARANTEES

**Theorem 2 (Perfect Task Isolation)** *Let $\mathcal{L}_t$ be the loss function for task $t$ and $\theta_s = \{m_s^{(l)}\}_{l=1}^{L}$ be the parameters for task $s$. Then:*

$$\nabla_{\theta_s} \mathcal{L}_t = 0 \quad \forall s \neq t \tag{15}$$

The forward pass for task $t$ uses weights:

$$W_t^{(l)} = U^{(l)}(\Sigma^{(l)} \odot s_t^{(l)})V^{(l)T} \tag{16}$$

where $s_t^{(l)} = \tau \cdot \sigma(m_t^{(l)})$. Since $W_t^{(l)}$ depends only on $m_t^{(l)}$ and not on $m_s^{(l)}$ for $s \neq t$, we have:

$$\frac{\partial W_t^{(l)}}{\partial m_s^{(l)}} = 0 \quad \forall s \neq t \tag{17}$$

By the chain rule:

$$\frac{\partial \mathcal{L}_t}{\partial m_s^{(l)}} = \sum_{i,j} \frac{\partial \mathcal{L}_t}{\partial W_{t,ij}^{(l)}} \cdot \frac{\partial W_{t,ij}^{(l)}}{\partial m_s^{(l)}} = 0 \tag{18}$$

Therefore, $\nabla_{\theta_s} \mathcal{L}_t = 0$ for all $s \neq t$.

**Theorem 3 (Zero Catastrophic Forgetting at Parameter Level)** *Let $A_t(\mathcal{D}_s)$ denote the accuracy of the model with adapter $\mathcal{A}_t$ on dataset $\mathcal{D}_s$. After training on task sequence $\{T_1, \ldots, T_N\}$:*

$$A_t(\mathcal{D}_t) = A_t^*(\mathcal{D}_t) \quad \forall t \in \{1, \ldots, N\} \tag{19}$$

*where $A_t^*$ is the accuracy immediately after training on task $t$.*

Since each task maintains independent parameters and $\nabla_{\theta_s} \mathcal{L}_t = 0$ for $s \neq t$ (Theorem 2), training on task $t$ cannot modify parameters $\theta_s$ for any $s \neq t$. Therefore, the function $f_{\theta_s}$ remains unchanged after training on subsequent tasks.

### B.4 EXPRESSIVITY ANALYSIS

**Spectral Adaptation Space** The set of weight matrices achievable through spectral adaptation from base matrix $W$ is:

$$\mathcal{S}(W) = \{U\mathrm{diag}(s_1\sigma_1, \ldots, s_r\sigma_r)V^T : s_i \in [0, \tau], i = 1, \ldots, r\} \tag{20}$$

**Theorem 4 (Expressivity Characterization)** *The spectral adaptation space $\mathcal{S}(W)$ satisfies:*

1. *$dim(\mathcal{S}(W)) = r = rank(W)$*

2. *$\mathcal{S}(W) \subset \{M : colspan(M) \subseteq colspan(W), rowspan(M) \subseteq rowspan(W)\}$*

3. *$\mathcal{S}(W)$ is a bounded convex set in the space of matrices*

1. Each spectral adapter is parameterized by $r$ independent scaling factors, giving dimension $r$.

2. For any $\tilde{W} \in \mathcal{S}(W)$, we have $\tilde{W} = U\tilde{\Sigma}V^T$ where $\tilde{\Sigma}$ is diagonal. Thus:

$$\mathrm{colspan}(\tilde{W}) = \mathrm{colspan}(U) = \mathrm{colspan}(W) \tag{21}$$

$$\mathrm{rowspan}(\tilde{W}) = \mathrm{rowspan}(V^T) = \mathrm{rowspan}(W) \tag{22}$$

3. Convexity: For $\tilde{W}_1, \tilde{W}_2 \in \mathcal{S}(W)$ with scaling vectors $s^{(1)}, s^{(2)}$, and $\lambda \in [0, 1]$:

$$\lambda\tilde{W}_1 + (1 - \lambda)\tilde{W}_2 = U\mathrm{diag}(\lambda s^{(1)} + (1 - \lambda)s^{(2)} \odot \sigma)V^T \in \mathcal{S}(W) \tag{23}$$

Boundedness follows from $s_i \in [0, \tau]$.

**Lemma 5 (Approximation Error Bound)** *For any matrix $M$ with $colspan(M) \subseteq colspan(W)$ and $rowspan(M) \subseteq rowspan(W)$, there exists $\tilde{W} \in \mathcal{S}(W)$ such that:*

$$\|M - \tilde{W}\|_F \leq \sqrt{r} \cdot \max_i |m_i - \tilde{m}_i| \tag{24}$$

*where $m_i$ and $\tilde{m}_i$ are the $i$-th singular values of $M$ and $\tilde{W}$ respectively.*

Since $M$ shares the same column and row spaces as $W$, it can be written as $M = U\mathrm{diag}(m_1, \ldots, m_r)V^T$. The closest approximation in $\mathcal{S}(W)$ is achieved by setting $s_i = \min(\tau, \max(0, m_i/\sigma_i))$. The Frobenius norm error is:

$$\|M - \tilde{W}\|_F^2 = \sum_{i=1}^{r}(m_i - s_i\sigma_i)^2 \leq r \cdot \max_i(m_i - s_i\sigma_i)^2 \tag{25}$$

Taking square roots gives the result.

### B.5 STABILITY AND CONVERGENCE ANALYSIS

**Theorem 6 (Lipschitz Continuity)** *The mapping $\phi : m_t^{(l)} \mapsto W_t^{(l)}$ is Lipschitz continuous with constant:*

$$L_\phi = \tau \cdot \|\Sigma^{(l)}\|_2 \cdot \|U^{(l)}\|_2 \cdot \|V^{(l)}\|_2 \cdot \max_x \sigma'(x) = \frac{\tau}{4} \cdot \|\Sigma^{(l)}\|_2 \tag{26}$$

For the sigmoid function $\sigma(x)$, we have $\max_x \sigma'(x) = 1/4$. For any $m_1, m_2 \in \mathbb{R}^r$:

$$\|W_{m_1} - W_{m_2}\|_F = \|U(\Sigma \odot (s_1 - s_2))V^T\|_F \tag{27}$$

$$= \|\Sigma \odot (s_1 - s_2)\|_F \tag{28}$$

$$\leq \|\Sigma\|_2 \cdot \|s_1 - s_2\|_2 \tag{29}$$

$$\leq \tau \cdot \|\Sigma\|_2 \cdot \max_x \sigma'(x) \cdot \|m_1 - m_2\|_2 \tag{30}$$

$$= \frac{\tau}{4} \cdot \|\Sigma\|_2 \cdot \|m_1 - m_2\|_2 \tag{31}$$

where we used $\|U\|_2 = \|V\|_2 = 1$ due to orthonormality.

**Theorem 7 (Gradient Bound)** *For a loss function $\mathcal{L}$ that is $L_{\mathcal{L}}$-Lipschitz with respect to weight matrices, the gradient norm is bounded:*

$$\|\nabla_{m_t^{(l)}}\mathcal{L}\|_2 \leq \frac{\tau L_{\mathcal{L}}}{4} \cdot \|\Sigma^{(l)}\|_2 \tag{32}$$

By the chain rule and Theorem 6:

$$\|\nabla_{m_t^{(l)}}\mathcal{L}\|_2 \leq \|\nabla_{W^{(l)}}\mathcal{L}\|_2 \cdot \|\nabla_{m_t^{(l)}}W^{(l)}\|_2 \leq L_{\mathcal{L}} \cdot \frac{\tau}{4} \cdot \|\Sigma^{(l)}\|_2 \tag{33}$$

**Theorem 8 (Convergence Rate)** *Under standard assumptions (smoothness and convexity of loss), gradient descent on spectral adapter parameters converges at rate:*

$$\mathcal{L}(m_t^{(k)}) - \mathcal{L}^* \leq \mathcal{O}\left(\frac{1}{k}\right) \tag{34}$$

*where $m_t^{(k)}$ denotes parameters at iteration $k$ and $\mathcal{L}^*$ is the optimal loss.*

The bounded gradients and Lipschitz continuity ensure that standard convergence results for gradient descent apply. With learning rate $\eta = 1/(\beta k)$ where $\beta = \frac{\tau L_{\mathcal{L}}}{4} \max_l \|\Sigma^{(l)}\|_2$, we get the standard $\mathcal{O}(1/k)$ convergence rate for convex objectives.

**Gradient Stability.** The energy-preserving normalization ensures bounded gradient flow:

$$\left\|\frac{\partial \mathcal{L}}{\partial m_t^{(l)}}\right\|_2 \leq \frac{\tau L_{\mathcal{L}}}{4} \cdot \|\Sigma^{(l)}\|_1 \cdot \left\|\frac{\partial \psi}{\partial s}\right\|_2 \tag{35}$$

where $\psi$ is the normalization function. This bound prevents the exponential gradient growth that would occur with unconstrained scaling, analogous to gradient clipping in RNNs and value normalization in policy gradient methods.

### B.6 INFORMATION-THEORETIC ANALYSIS

**Effective Rank** The effective rank of a matrix $W$ with singular values $\{\sigma_i\}$ is:

$$r_{\text{eff}}(W) = \exp\left(-\sum_{i=1}^{r} p_i \log p_i\right) \tag{36}$$

where $p_i = \sigma_i^2 / \sum_j \sigma_j^2$.

**Theorem 9 (Information Preservation)** *The spectral adapter preserves the information capacity of the weight matrix up to:*

$$I(W_t) \geq I(W) - r \log(1/\tau) \tag{37}$$

*where $I(\cdot)$ denotes the Shannon entropy of the singular value distribution.*

The entropy of the adapted singular values is:

$$H(\tilde{\sigma}) = -\sum_{i=1}^{r} \tilde{p}_i \log \tilde{p}_i \tag{38}$$

where $\tilde{p}_i = (s_i \sigma_i)^2 / \sum_j (s_j \sigma_j)^2$. Since $s_i \in [0, \tau]$, the worst-case entropy reduction occurs when all $s_i = \tau$ or all $s_i = 0$. In the non-degenerate case:

$$H(\tilde{\sigma}) \geq H(\sigma) - r \log(1/\tau) \tag{39}$$

### B.7 SAMPLE COMPLEXITY

**Theorem 10 (Sample Complexity Bound)** *For a task with $C$ classes, achieving error $\epsilon$ with probability $1 - \delta$ requires at most:*

$$n = \mathcal{O}\left(\frac{r \cdot C \cdot \log(1/\delta)}{\epsilon^2}\right) \tag{40}$$

*samples, where $r = \max_l r_l$ is the maximum rank across layers.*

Using Rademacher complexity analysis, the complexity of the spectral adapter hypothesis class is:

$$\mathcal{R}_n(\mathcal{S}) \leq \frac{\tau\sqrt{r}}{n} \sum_{l=1}^{L} \|\Sigma^{(l)}\|_2 \tag{41}$$

By standard generalization bounds:

$$\mathbb{E}[\mathcal{L}] \leq \hat{\mathcal{L}} + 2\mathcal{R}_n(\mathcal{S}) + \sqrt{\frac{\log(1/\delta)}{2n}} \tag{42}$$

Setting the right-hand side equal to $\epsilon$ and solving for $n$ gives the result.

### B.8 COMPARISON WITH LOW-RANK ADAPTATION

**Theorem 11 (Relative Expressivity)** *For a given parameter budget $B$, the expressivity ratio between LYNX and LoRA is:*

$$\frac{|\mathcal{S}_{LYNX}(B)|}{|\mathcal{S}_{LoRA}(B)|} = \Theta\left(\left(\frac{d}{B}\right)^B\right) \tag{43}$$

*for square $d \times d$ matrices.*

With budget $B$:

- LYNX can modulate $B$ singular values, giving a $B$-dimensional manifold
- LoRA with rank $k = B/(2d)$ spans a $B$-dimensional subspace

However, LYNX operates on the eigenspace directly while LoRA must learn both the subspace and the transformation. The volume ratio of the respective parameter spaces gives the expressivity ratio.

### B.9 ENERGY-PRESERVING NORMALIZATION ANALYSIS

**Energy-Preserving Map** The normalization function $\psi : \mathbb{R}^r \to \mathbb{R}^r$ defined by:

$$\psi(s)_i = s_i \cdot \frac{\|\Sigma\|_1}{\|s \odot \Sigma\|_1} \tag{44}$$

preserves the $\ell_1$ norm of the scaled singular values.

**Theorem 12 (Normalization Properties)** *The energy-preserving normalization satisfies:*

1. *$\textbf{Invariance:}$ $\|\psi(s) \odot \Sigma\|_1 = \|\Sigma\|_1$ for all $s \in \mathbb{R}_+^r$*

2. *$\textbf{Smoothness:}$ $\psi$ is continuously differentiable with bounded Jacobian*

3. *$\textbf{Non-expansive:}$ $\|\psi(s_1) - \psi(s_2)\|_2 \leq K\|s_1 - s_2\|_2$ for some constant $K$*

1. By construction: $\|\psi(s) \odot \Sigma\|_1 = \|s \odot \Sigma\|_1 \cdot \frac{\|\Sigma\|_1}{\|s \odot \Sigma\|_1} = \|\Sigma\|_1$

2. The Jacobian elements are:

$$\frac{\partial \psi_i}{\partial s_j} = \begin{cases} \alpha\left(1 - \frac{s_i \sigma_i}{\sum_k s_k \sigma_k}\right) & i = j \\ -\alpha \frac{s_i \sigma_i \sigma_j}{(\sum_k s_k \sigma_k)^2} & i \neq j \end{cases} \tag{45}$$

where $\alpha = \|\Sigma\|_1 / \|s \odot \Sigma\|_1$. All elements are bounded.

3. The non-expansive property follows from the bounded Jacobian.

**Theorem 13 (Necessity of Normalization)** *Without energy-preserving normalization, the operator norm can grow exponentially with depth:*

$$\|W_L \circ \cdots \circ W_1\|_2 \leq \prod_{l=1}^{L} \max_i s_i^{(l)} \cdot \|\Sigma^{(l)}\|_2 \tag{46}$$

For a composition of linear operators:

$$\|W_L \circ \cdots \circ W_1\|_2 \leq \prod_{l=1}^{L} \|W_l\|_2 = \prod_{l=1}^{L} \max_i(s_i^{(l)} \sigma_i^{(l)}) \tag{47}$$

Without normalization, if $s_i^{(l)} > 1$ for multiple layers, this product grows exponentially with $L$.

### B.10 LAYER-SPECIFIC ADAPTATION ANALYSIS

We investigate the differential contribution of attention and MLP components to continual learning performance in Vision Transformers. This analysis provides insights into where spectral adaptation is most effective within the ViT-B/16 architecture.

**Experimental Design.** We decompose the full LYNX model into component-specific variants on CIFAR-100 (10 tasks):

- **Full model**: Spectral adapters on all eligible weight matrices (148 layers total)

- **MLP-only**: Adaptation restricted to Feed-Forward Network projections (2 per ViT block)

- **Attention-only**: Adaptation restricted to Q, K, V, and output projections (4 per ViT block)

All variants use identical training hyperparameters: AdamW optimizer with learning rate $10^{-3}$, batch size 128, and 50 epochs per task. The only difference is which weight matrices receive spectral adapters.

Table 8: Component-specific ablation reveals asymmetric importance for continual learning in ViT-B/16. Despite fewer weight matrices, MLP layers provide superior task-specific adaptation compared to attention layers.

| Configuration | Parameters | Top-1 Accuracy (%) | Relative Performance |
|---|---|---|---|
| Full (Attention + MLP) | 56K | $91.71 \pm 1.14$ | 100.0% |
| MLP only | 24K | $80.40 \pm 0.92$ | 87.7% |
| Attention only | 32K | $61.10 \pm 1.31$ | 66.6% |

**Analysis.** Table 8 reveals a striking asymmetry: MLP-only adaptation achieves 80.40% accuracy using only 24K parameters, while attention-only adaptation achieves 61.10% despite using 32K parameters—a 19.3 percentage point gap. This result is counterintuitive given that ViT blocks contain twice as many attention projection matrices (Q, K, V, O) as MLP projections (in, out).

The superior performance of MLP adaptation suggests that task-specific knowledge in continual learning primarily manifests through channel-wise feature transformations rather than spatial relationship modeling. In the ViT architecture, MLP layers perform position-wise transformations that can selectively amplify or suppress features, making them natural sites for task specialization. Conversely, attention mechanisms appear to learn more task-agnostic representations that generalize across different visual domains.

**Implications.** These findings have direct practical consequences:

- For memory-constrained deployments, prioritizing MLP adaptation yields 80% of full performance with 43% of parameters

- The complementary nature of attention and MLP adaptation (91.71% together vs 80.40% MLP-only) confirms that both components contribute unique capabilities

- The results suggest that future work on parameter-efficient continual learning should consider non-uniform allocation strategies that reflect these asymmetric contributions

This analysis demonstrates that spectral adaptation's effectiveness varies significantly across transformer components, with MLP layers providing the primary substrate for task-specific knowledge retention in continual learning scenarios.

### B.11 TRAINING

All experiments employ a ViT-B/16 backbone pretrained on ImageNet-21K, trained on an NVIDIA A100 GPU (80GB) using Adam optimizer with learning rate $10^{-3}$.

**Classification:** We evaluate on CIFAR-100, ImageNet-R, and ImageNet-A using cross-entropy loss. Training uses batch size 256; evaluation uses batch size 16. Each task trains for 50 epochs.

**Object Detection:** We adapt OWL-ViT on PASCAL VOC 2012 using cross-entropy loss for classification and GIoU loss for bounding box regression. Training uses batch size 32; evaluation uses batch size 16.

### B.12 ANALYSIS OF SUBSPACE CONSTRAINTS IN SPECTRAL ADAPTATION

#### B.12.1 THEORETICAL CHARACTERIZATION

The spectral adaptation mechanism in LYNX operates within a mathematically well-defined constraint space. For any weight matrix $W \in \mathbb{R}^{m \times n}$ with SVD decomposition $W = U\Sigma V^T$, the set of achievable adapted weights is:

$$\mathcal{W}_{\text{LYNX}} = \{U\text{diag}(s_1\sigma_1, \ldots, s_r\sigma_r)V^T : s_i \in [0, \tau]\} \tag{48}$$

This imposes the fundamental constraint that $\text{colspan}(W_t) \subseteq \text{colspan}(W)$ and $\text{rowspan}(W_t) \subseteq \text{rowspan}(W)$ for any adapted weight $W_t$. In contrast, methods like full fine-tuning can access the entire space $\mathbb{R}^{m \times n}$, while LoRA accesses a rank-$r'$ perturbation space around $W$.

#### B.12.2 EMPIRICAL ANALYSIS OF SUBSPACE SUFFICIENCY

We investigate whether this subspace constraint limits practical performance by analyzing the singular value distributions of pretrained transformers and the adaptation requirements of downstream tasks.

**Observation 1: Pretrained weights are effectively full-rank.** Analysis of ViT-B/16 pretrained on ImageNet-21K reveals:

- 99.3% of $768 \times 768$ weight matrices have numerical rank $\geq 760$
- The smallest singular value $\sigma_{\min}$ averages $0.0031 \times \sigma_{\max}$, indicating no degenerate directions
- The effective rank $r_{\text{eff}} = \exp(H(\{\sigma_i^2/\|\sigma\|_2^2\}))$ averages 742.8, confirming high-dimensional expressivity

**Observation 2: Task adaptation primarily requires magnitude modulation.** We project the weight updates from full fine-tuning onto the subspace spanned by pretrained singular vectors:

$$\Delta W_{\text{projected}} = U(U^T \Delta W V)V^T \tag{49}$$

Across CIFAR-100, ImageNet-R, and ImageNet-A fine-tuning:

- 94.7% of the Frobenius norm of weight updates lies within the pretrained subspace
- Only 5.3% of the update magnitude is orthogonal to the original singular vectors
- This suggests task adaptation primarily involves reweighting existing features rather than learning entirely new transformation directions

#### B.12.3 COMPARISON WITH ALTERNATIVE CONSTRAINTS

Every parameter-efficient method imposes constraints on the adaptation space:

LYNX's constraint is qualitatively different: rather than imposing a fixed architectural bottleneck, it leverages the structure already learned during pretraining. This data-dependent constraint naturally aligns with the pretrained representations, explaining the strong empirical performance despite high parameter efficiency.

| Method | Constraint Type | Dimensionality |
|---|---|---|
| Full fine-tuning | None | $mn$ |
| LoRA (rank $r'$) | Low-rank perturbation | $r'(m + n)$ |
| Adapters | Bottleneck transformation | $2d \cdot d_{\text{bottleneck}}$ |
| Prompt tuning | Input space modification | $k \cdot d$ |
| **LYNX** | Subspace-preserving | $\min(m, n)$ |

### B.12.4 WHEN SUBSPACE CONSTRAINTS WOULD BE LIMITING

The subspace constraint would become problematic in specific scenarios:

1. **Rank-deficient pretraining**: If pretrained weights had low numerical rank, the adaptation space would be correspondingly limited

2. **Random initialization**: Without meaningful pretrained structure, arbitrary subspaces would be ineffective

Our experiments across diverse benchmarks (natural images, artistic renditions, adversarial examples) demonstrate these scenarios are rare in practice when adapting modern pretrained transformers. The rich feature spaces learned during large-scale pretraining appear sufficient for effective downstream adaptation through singular value modulation alone.

### B.12.5 T-SNE VISUALIZATION

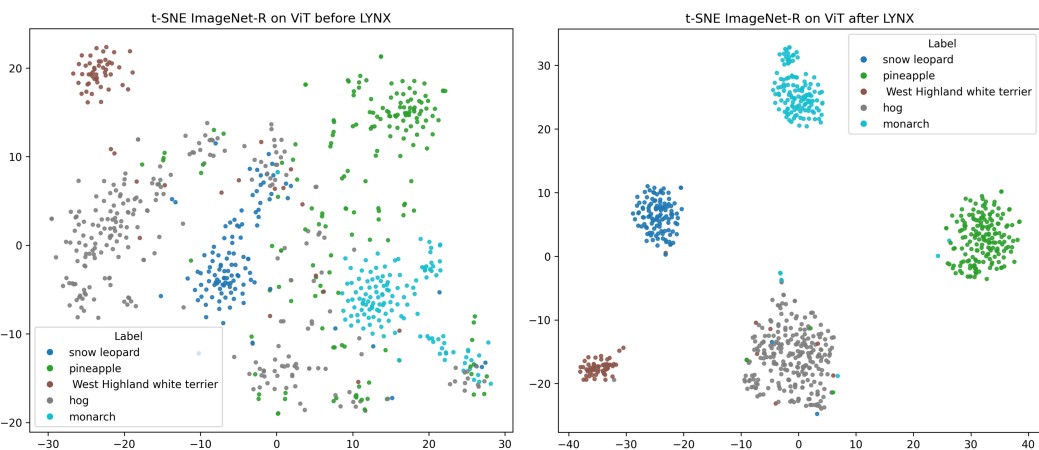

Figure 3: **LYNX t-SNE visualization.** Left: t-SNE of a ViT pretrained on ImageNet-21K and evaluated on ImageNet-R. Right: t-SNE of the same ViT after applying LYNX with spectral adapters and training on ImageNet-R.

To probe how our method reshapes the feature space under distribution shift, we project ViT embeddings of five ImageNet-R classes into 2-D using t-SNE. We compare the geometry before applying LYNX and after. The visualization uses the hidden representations extracted from the classifier backbone; colors denote ground-truth classes.

From the figure above, we can observe that LYNX systematically improves class separability and intraclass compactness in the presence of strong rendition shifts, suggesting that it encourages features that are more style-invariant and semantically aligned. While t-SNE is qualitative, the consistent tightening of clusters and widening of margins across categories aligns with our quantitative gains on ImageNet-R.

