# OpenReview forum: "LYNX - Lightweight Yielding Network eXpansion"
_ICLR.cc/2026/Conference — ICLR 2026 Conference Withdrawn Submission_

### Official Review · Reviewer_bh4y · 2025-10-30

**Soundness:** 2
**Presentation:** 2
**Contribution:** 2
**Rating:** 2
**Confidence:** 3

**Summary:**

This paper proposes a parameter efficient method for class incremental continual learning. The approach relies on SVD decomposition of model weights, learning a set of scaling parameters per task to rescale singular values. Parameter selection is carried out either based on task knowledge at inference time, or by prediction confidence score. It is unclear whether experiments rely on task knowledge at test time.  Experiments show strong performance for classification tasks with respect to competing methods, and show that it can be used in detection settings as well.

**Strengths:**

Spectral adaptation is a promising idea for parameter efficient fine-tuning, which has demonstrated great parameter efficiency and ability to adapt models successfully. The idea of leveraging this for continual adaptation is good, as dedicated parameters can be learned and stored cheaply.

The paper is easy to read and follow for the most part. The properties and benefits of the methods are carefully analysed, although important experiments and ablations are only available in supplementary materials (e.g. influence of the choice of layers to adapt).
Experiments show that good results on classification tasks, outperforming other methods, and authors compare to a large array of alternative methods.

**Weaknesses:**

My main issue with the paper is its limited novelty. Spectral adaptation of models is not new, and has been proven to be a powerful and efficient adaptation technique in prior works [1,2].
The key innovation seems to be the use of this technology in the continual learning setting. While this can be a valuable contribution, and I appreciate that the authors are careful to always highlight that this is the first use of this method in the context of continual learning, the paper is written in a way that can lead to reader to believe this work is the first spectral adaptation based approach. Except for a brief mention of LoRA-XS in the related works section, this line of work is not acknowledged in the introduction, nor in the methodological description. Key inspirations should be clearly acknowledged, so that innovations of this work can be properly highlighted. This lack of grounding in the literature makes it more challenging to highlight what innovations were necessary to leverage the PEFT method in a continual learning setting.
Based on the paper in its current form, innovation efforts seem relatively limited, with a normalisation step and a task selection procedure at inference time. I have not been able to find an indication that the latter is used in following experiments in the paper. As such, this approach seems to be a nearly direct application of pre-existing technology to the class incremental problem.
Regarding experiments, it is appreciated that the authors have made the effort to report results in a different scenario (object detection). However, providing results solely for the proposed method is not very informative or useful. Providing simple baselines and references (such as an oracle, or a regular sequential fine-tuning baseline) would help ground results better. Class incremental object detection has also been heavily studied, and alternative methods could be reported for context.

[1] SVDiff: Compact Parameter Space for Diffusion Fine-Tuning, Han et al, ICCV 2023
[2] SVFT: Parameter-Efficient Fine-Tuning with Singular Vectors, Lingam et al, NeurIPS 2024

**Questions:**

-	Ultimately, I see a lot of potential in the ideas discussed in this work and using very parameter efficient techniques for continual learning. However, the paper in its current form needs much better grounding in relevant pre-existing PEFT literature to clarify novelty and claims. As the focus of this work is spectral adaptation for continual learning, it is important to highlight the challenges of adapting to this particular setting, and what key contributions are provided here.
-	The singular value scaling seems to differ from prior work by multiplying the learnable weights instead of an additive approach. This seems to lead to more instability, requiring normalisation. Why was this choice made vs pre-existing works such as SVFT?
-	It is mentioned that only this approach can allow complete task isolation. Wouldn’t lora models be able to do that as well, dedicating a separate set of weights to different tasks? While this would be substantially more parameter expensive, it would allow complete parameter isolation. My apologies if I misunderstood the claim.
-	How are task managed at inference time in experiments? Is task knowledge known or is the confidence based approach used? How does the inference mode affect performance?
-	It is mentioned that achieving high score for high granularity tasks should be more challenging, however, higher granularity tasks with dedicated parameters would simplify the problem substantially as classification needs to be done across fewer tasks.
-	Please make sure to proof read the paper carefully before submission. There are typos (I assume it is “task” and not “mask” in the abstract), a missing reference (line 63), and references should use \citep for correct readability.

---

### Official Review · Reviewer_Xvcm · 2025-10-30

**Soundness:** 4
**Presentation:** 4
**Contribution:** 4
**Rating:** 6
**Confidence:** 3

**Summary:**

The authors proposed the Lightweight Yielding Network eXpansion (LYNX), which adopted the concept of SVD to decompose weights into orthogonal directions (singular vectors) and corresponding magnitudes (singular values), and only learned the scaling of singular values, which highly reduced the computational cost. LYNX also showed performance gains on other PEFT methods across various image classification and object detection datasets with minimal overhead.

**Strengths:**

1. The method is technically sound and insightful.
2. The method performs well on several domains: standard image classification, distribution shift classification, and object detection.

**Weaknesses:**

The proposed method might have some limitations, although the authors conduct empirical analyses claiming that LYNX can work in most scenarios, unless the weight is either randomly initialized or of low numerical rank during pretraining. I'd like to argue that it's not that rare to encounter a low numerical rank weight. It is unclear that the method would be effective on models that have already SFT or RFT. Although the authors evaluated on some out-of-domain data such as ImageNet-A and an object detection dataset, it would be better if we could test on more extreme OOD datasets such as medical data, satellite data, and industrial data, etc.

In Appendix B.12.2, the authors observe that for ViT-B/16 pretrained on ImageNet-21K, 99.3% of 768x768 weight matrices have a numerical rank \>= 760, which contradicts the claim at line 82 "For typical transformer architectures where r ≪ d, this yields orders of magnitude fewer parameters than methods that scale with hidden dimensions." Please clarify it.

Lastly, the authors should discuss with works that fine-tune with SVD such as [1].

[1] Lingam, Vijay Chandra, et al. "Svft: Parameter-efficient fine-tuning with singular vectors." Advances in Neural Information Processing Systems 37 (2024): 41425-41446.

**Questions:**

My questions are at the weakness part, it’s not clear how LYNX performs on different settings such as OOD dataset and architecture that is low-rank. The author should conduct some experiments on such settings to show the limitations.

Minor Suggestions
1. At line 63, the work is not correctly cited
2. The text size of Figure 1 is too small.

---

### Official Review · Reviewer_tsLZ · 2025-10-31

**Soundness:** 2
**Presentation:** 2
**Contribution:** 2
**Rating:** 4
**Confidence:** 3

**Summary:**

The paper proposes LYNX, a continual learning method that first factorizes each layer with SVD and then learns tiny task adapters by scaling singular values while keeping the factors frozen. The adapters are kilobyte-scale and can be swapped per task. This design limits memory growth and reduces interference because the backbone stays fixed and only a few parameters change per task. It supports both task-aware and simple task-free use at inference with little overhead. Experiments on standard class-incremental benchmarks show competitive or better accuracy than strong PEFT baselines while using far fewer parameters. The method works on both ConvNets and ViTs and scales well as tasks increase.

**Strengths:**

- Parameter efficiency. The proposed “decompose-then-finetune” design keeps the backbone frozen and only learns tiny, swappable singular modulation vectors. This gives low memory growth per task, near-zero runtime overhead, and fast task switching.

- Competitive CL results across heterogeneous vision benchmarks. They report competitive average accuracy on long class-incremental classification tasks and detection task.

- The proposed energy preserving scheme is shown to be effective in regularizing the optimization.

**Weaknesses:**

- What’s the intuition behind this simple singular modulation design that it would work well for continual learning scenarios, specifically, for plasticity?

- Lack of comparison with previous decompose-then-finetune CL works [1,2]. The authors position their contribution as “kilobyte-scale, swappable adapters”. Yet, [2] also decomposes weights over a tiny set of tunable parameters with the term ‘atom swapping’. Meanwhile, [1] also proposes an efficient and effective expansion-based CL method that learns a rank-1 mask over each parameter matrix for each task. The authors should include discussions and comparisons with these works.

- Experimental setting is confusing. In Table 2, why the number of tasks is always changing, making the results hard to interpret. It also raises concerns on fair comparisons. In table 1, comparison with CACL should be included.

References:
1.	Wen, Y., Tran, D., & Ba, J. BatchEnsemble: an Alternative Approach to Efficient Ensemble and Lifelong Learning. In International Conference on Learning Representations.
2.	Miao, Z., Wang, Z., Chen, W., & Qiu, Q. Continual learning with filter atom swapping. In International Conference on Learning Representations.

**Questions:**

For 4-dim convolutional weight, how do you convert it to 2-dimension and conduct SVD?

---

### Official Review · Reviewer_UDVk · 2025-11-01

**Soundness:** 2
**Presentation:** 2
**Contribution:** 2
**Rating:** 2
**Confidence:** 5

**Summary:**

This paper proposes LYNX, a parameter-efficient continual learning method that adapts neural networks by modulating only the singular values of pre-decomposed weight matrices, while keeping the singular vectors fixed. Each task learns a small scaling vector, enabling compact task adapters. Although the method achieves strong empirical results and clear presentation, it mainly extends prior SVD-based fine-tuning techniques to the continual learning setting, with limited novelty. Theoretical claims about “perfect task isolation” rely on known task IDs and do not truly address forgetting in task-free scenarios, and the evaluation protocol deviates from standard class-incremental learning.

**Strengths:**

1. The paper is clearly written and easy to follow.

2. LYNX is highly parameter-efficient, requiring only small task-specific scaling vectors.

**Weaknesses:**

1. Limited methodological novelty.
The core idea of adapting networks by adjusting only singular values has been previously explored in SVFT: Parameter-Efficient Fine-Tuning with Singular Vectors. LYNX essentially extends this concept to continual learning but introduces little conceptual novelty beyond task-specific scaling.

2. Unconvincing explanation for forgetting prevention.
It remains unclear why modifying only singular values should inherently prevent forgetting.
Although Theorem 2 (“Perfect Task Isolation”) shows that gradients for different task parameters are independent, this holds only when task identities are explicitly known—each task has its own adapter.
In practice, without task IDs, catastrophic forgetting can still occur.
Furthermore, fixing the singular vectors might restrict adaptation when gradient directions are not aligned with the SVD basis.

3. Questionable evaluation protocol.
As stated in line 336, “we measure Top-1 accuracy on each task’s test set independently.”
This deviates from the standard CIL  setting, where the model must jointly classify among all previously seen classes.
Consequently, the reported accuracies may overestimate real CIL performance.
In Table 2, comparisons across methods are inconsistent—different numbers of tasks (5, 10, 20, 40) and even different metrics (Top-1 vs. Top-5) make the results hard to interpret and potentially unfair.
Larger task counts reduce the number of classes per task, which naturally inflates per-task accuracy, hence the comparison between different task counts may not be meaningful.

4. Practical limitations of task-free inference.
The “task-free inference” mode requires evaluating all task adapters and selecting the one with the highest confidence, resulting in inference cost proportional to the number of tasks.
This is computationally impractical for real deployments.
Besides, the paper does not clearly indicate which experiments use task-aware vs. task-free inference, making it difficult to assess the true effectiveness under realistic conditions.

**Questions:**

see weekness

---

### Note · Authors · 2026-01-26

I have read and agree with the venue's withdrawal policy on behalf of myself and my co-authors.

---

### Meta-Review · Area_Chair_w17K · 2026-01-06

**Summary:**

Reviewers recognize LYNX as a parameter-efficient continual learning method with competitive empirical results. However, most reviewers raise significant concerns about limited conceptual novelty, unclear positioning relative to prior SVD/PEFT work, and arguable evaluation protocols.

**Reviewer Concerns:**

There was no rebuttal, so the above concerns remain unaddressed.

**Reviewer Scores:**

Overall, the majority of the reviews lean towards rejection. In the absence of a rebuttal or discussion, reviewer scores were unlikely to change.

---

### Decision · Program_Chairs · 2026-01-26

Reject